# A Case of Oncocytic Adrenal Cortical Neoplasm with Uncertain Malignant Potential Turned Out to Be Oncocytic Adrenal Cortical Carcinoma with Distant Metastasis: Could Pathology Do Better Initially?

**DOI:** 10.3390/medicina58070900

**Published:** 2022-07-06

**Authors:** Chien-Peng Huang, Ming-Shen Dai, Chien-Chang Kao, Wen-Chiuan Tsai, Cheng-Ping Yu

**Affiliations:** 1Department of Pathology, Tri-Service General Hospital, National Defense Medical Center, Taibei 11490, Taiwan; bunnypeng0529@gmail.com (C.-P.H.); ab95057@hotmail.com (W.-C.T.); 2Division of Hematology, Tri-Service General Hospital, National Defense Medical Center, Taibei 11490, Taiwan; dms1201@gmail.com; 3Division of Genitourinary, Tri-Service General Hospital, National Defense Medical Center, Taibei 11490, Taiwan; guman2011@gmail.com

**Keywords:** adrenal oncocytic neoplasms, oncocytic adrenal cortical neoplasm with uncertain malignant potential, oncocytic adrenal cortical carcinoma

## Abstract

Oncocytic adrenal cortical neoplasms are rare cases and are divided into oncocytoma, oncocytic neoplasms of uncertain malignant potential and oncocytic adrenal cortical carcinomas, based on the Lin–Weiss–Bisceglia (LWB) histological system adopted in the current World Health Organization (WHO). We reported a 42-year-old female diagnosed with an oncocytic neoplasm of uncertain malignant potential initially, which turned out to be a carcinoma owing to distant metastasis to the scalp and lung. To our knowledge, this is the first published case of oncocytic adrenal cortical carcinoma with scalp metastasis. This case also highlights the limitation of the current diagnostic algorithm and emphasizes the importance of two parameters (PHH3 and Ki-67) for determining the malignant potential of oncocytic adrenal cortical neoplasms.

## 1. Introduction

Adrenal cortical carcinoma (ACC) is the most common primary cancer in the adrenal gland and is subclassified into four histologic variants: conventional, oncocytic, myxoid, and sarcomatoid. The majority is the conventional variant, which accounts for 90%, followed by the oncocytic variant.

“Oncocyte” was first introduced in 1931 by Hamperl, referring to a cell with abundant granular eosinophilic cytoplasms, which are confirmed by an abundance of mitochondria ultrastructurally or immunohistochemically reactive to antimitochondrial antibody (AMA) mES-13 [1]. Oncocytic tumors frequently occur in the salivary glands, kidneys, thyroid, parathyroids, and pituitary gland. Oncocytic neoplasms rarely arise within the adrenal glands. According to the definition, these tumors should be composed of at least 50% oncocytic cells, which could further be classified as a mixed (50–90% oncocytic cells) or pure (>90%) variant [2].

Adrenal oncocytic neoplasms (AONs) are an extremely rare variant of adrenocortical tumors that include adrenal cortical oncocytoma, oncocytic adrenal cortical neoplasm with uncertain malignant potential, and oncocytic adrenal cortical carcinoma. Since the first case was confirmed by electron microscopy in 1991, only 56 cases were diagnosed as oncocytic adrenal cortical carcinoma until 2021 [3].

Based on current limited cases, the prognosis of oncocytic ACC appears to be better than for the conventional type. Only seven cases reported developed distant metastases, and the liver, lung, bone, and ovary were the most common sites for them (Table 1) [3].

## 2. Case Report

A 42-year-old female with a history of asthma without medication control came to our emergency department owing to diffuse abdominal pain in June 2020. The physical examination revealed epigastric and left upper quadrant tenderness as well as left costovertebral angle tenderness. The laboratory examination results showed as follows: Cortisol: 9.32 ug/dL (5 to 25 ug/dL), Testosterone: 26 ng/dL (4.94–32.01 nmol/L), Aldosterone: 56.5 pg/mL (20–90 pg/mL), P.R.A.: <0.14 ng/mL/h (1.02 ± 0.12 ng/mL/h), V.M.A.: 2.2 mg/24 h (2–7 mg/24 h). Abdominal CT revealed a heterogeneous low-density mass (size: 3.9 × 2.9 cm) over the left supra-renal space, accompanied with a low attenuation area in the center, suggestive of necrosis (Figure 1a). A laparoscopic adrenalectomy was performed.

Grossly, this yellow tumor measured 3.9 × 2.9 × 2.5 cm in size with a brownish central necrosis, smooth outer surface and intact adrenal capsule (Figure 1b). Microscopically, the tumor cells were characterized by abundant eosinophilic and vacuolated cytoplasm (Figure 2a) arranged in a trabecular growth pattern. The nuclei were vesicular with mild cytologic atypia. Mitoses were indistinct, and no atypical mitosis was found. Sinusoidal invasion (Figure 2b) and necrosis were seen. The differential diagnosis based on the location and hematoxylin and eosin stain included oncocytic pheochromocytoma, malignant eosinophilic neoplasm from the kidney, adrenal cortical carcinoma, epithelioid gastrointestinal stromal tumor (GIST), perivascular epithelioid cell tumor (PEComa), alveolar soft part sarcoma, and metastatic hepatocellular carcinoma. Immunohistochemically, tumor cells were positive for alpha-inhibin, synaptophysin, and Melan-A; and negative for cytokeratin AE1/AE3, chromogranin-A, MART1, human melanoma black 45 (HMB-45), PAX 8, cathepsin K, and c-kit. Electron microscopy ultrastructurally displayed abundant mitochondria (Figure 2c), and the immunohistochemical stain of AMA mES-13 also showed a strong, diffuse and finely granular pattern (Figure 2d). Owing to the presence of two minor criteria, sinusoidal invasion and necrosis, but the lack of any major criteria (>5 mitoses/50 HPF, atypical mitosis, venous invasion), the initial pathological diagnosis was interpreted as oncocytic adrenal cortical neoplasm with an uncertain malignant potential based on current Lin–Weiss–Bisceglia (LWB) criteria proposed for AONs.

In August 2021, the patient came back to our hospital due to a small nodule found over her right scalp. The image examination revealed a 1.17 cm soft tissue mass at the right frontal scalp, and the following PET scan revealed several sub-centimeter nodules throughout the lung field. Therefore, a surgical resection of this scalp tumor and wedge resection of pulmonary nodules were performed. The pathology of both the scalp and lung tumors showed metastatic oncocytic adrenal cortical carcinoma, confirmed by a panel of immunohistochemistry.

## 3. Discussion

It can be difficult to diagnose the oncocytic variant of ACC, even when helped by ancillary tests. Krishnamurthy et al. shared a similar opinion, suggesting that the only unquestionable criterion of malignancy in an AON is the presence of metastasis or invasion (capsular and/or vascular) [6].

The Weiss criteria for malignancy in ACC cannot be directly applied to AONs because of the consistent presence of eosinophilic cytoplasm, a diffuse growth pattern, and a high nuclear grade [9]. AONs are usually classified by a Lin–Weiss–Bisceglia (LWB) histological system that consists of three major criteria: mitotic rate > 5 mitoses per 50 high power fields, any atypical mitosis, and venous invasion; and four minor criteria: >10 cm or >200 g, necrosis, capsular invasion, and sinusoidal invasion [10]. AONs without any major or minor criteria are classified as benign (adrenal cortical oncocytoma); tumors with only minor criteria are classified as borderline (oncocytic adrenal cortical neoplasm with uncertain malignant potential); tumors with any major criteria are classified as malignant (oncocytic adrenal cortical carcinoma) [9,11].

This case was initially reported as an oncocytic adrenal cortical neoplasm with an uncertain malignant potential based on the Lin–Weiss–Bisceglia criteria, owing to the presence of sinusoidal invasion and necrosis but the lack of any major criteria.

Unfortunately, distant scalp and pulmonary metastases were found about one year later. Owing to the rapid progression of the disease, we wondered whether this AON could be diagnosed as a malignancy the first time.

Multifactorial diagnostic schemes have been proposed to aid in the diagnosis of ACCs [10]. The Helsinki score published by a European group in 2015 relies on the mitotic rate, necrosis and Ki-67 index (Figure 3a) (3 × mitotic count [>5/50 high power fields] + 5 × presence of necrosis + Ki-67 proliferative index in the most proliferative spot of the tumor) of ACC and focuses on the predictive diagnosis as well as prognosis of any type of ACCs. A Helsinki score > 8.5 is associated with metastatic potential and warrants the diagnosis of ACC [3]. The Helsinki score of this present case will be 18 (3 + 5 + 10 = 18), which supports the diagnosis of oncocytic ACC.

Phospho-histone H3 (PHH3) immunostaining is a valid, accurate, and reproducible surrogate marker of the standard mitotic count in adrenal cortical tumors, especially in ACCs with a low mitotic index [12]. PHH3 is recommended for low-proliferating tumors to avoid an underestimation of the ‘mitotic figure parameter’ of Weiss and other diagnostic systems in a review article. Hence, we applied PHH3 immunostaining (Figure 3b) to look for hotspots and obtained a mitotic figure of more than five mitoses per 50 high power fields in one of these hotspots, which met one of the major criteria of the Lin–Weiss–Bisceglia histological system for diagnosing oncocytic ACC.

## 4. Conclusions

In the current WHO classification, the Lin–Weiss–Bisceglia (LWB) criteria proposed in 2004 are adopted to differentiate between AONs that are benign, borderline malignant potential, or malignant neoplasms. However, several articles have emphasized the importance of the Ki-67 proliferative index as a parameter for predicting diagnosis as well as prognosis. We provided a case which was initially diagnosed as an oncocytic adrenal cortical neoplasm with an uncertain malignant potential following LWB criteria. However, it turned out to be a malignancy owing to distant scalp and lung metastases about one year later. After reevaluating the mitotic count and proliferative index in hotspots by PHH3 and Ki-67 immunostaining, this case could be initially diagnosed as oncocytic ACC by either the Lin–Weiss–Bisceglia histological system or the Helsinki score. Hence, we would like to share this extremely rare case and emphasize the importance of PHH3 (identifying mitotic figures) and Ki-67 (proliferative index) in hotspots as ancillary tests for determining the malignant potential of AONs.

## Figures and Tables

**Figure 1 medicina-58-00900-f001:**
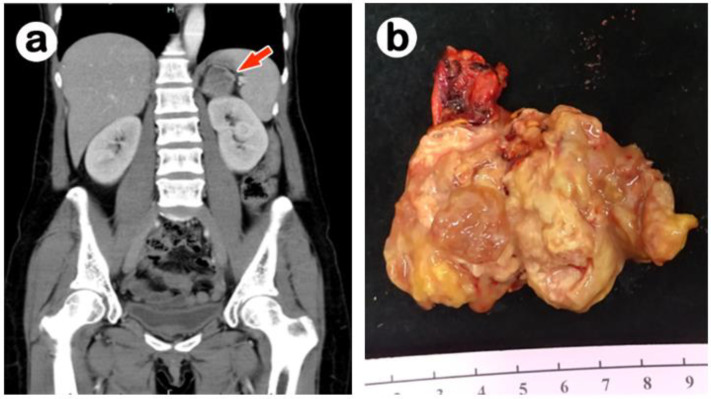
(**a**) Abdominal CT scan demonstrating the horizontal section of a heterogeneous low-density tumor in the left adrenal gland (red arrow). (**b**) The yellowish adrenal tumor grossly showed a focal brownish necrosis, smooth outer surface and intact capsule.

**Figure 2 medicina-58-00900-f002:**
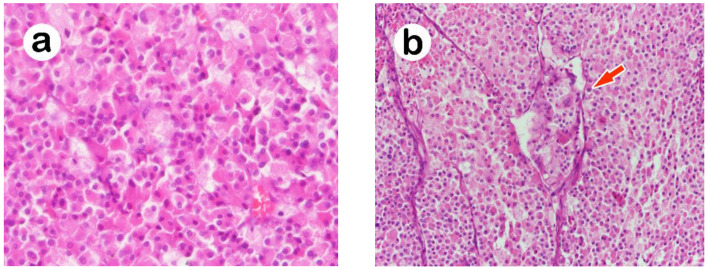
Oncocytic adrenal cortical neoplasm with uncertain malignant potential composed of tumor cells with (**a**) eosinophilic granular and occasional vacuolated clear cytoplasms (H&E 200×), arranged in (**b**) a trabecular pattern with sinusoidal invasion (red arrow) (H&E 200×). Electron microscopy showed (**c**) many mitochondria, further confirmed by (**d**) an immunohistochemical stain of AMA, mES-13, revealing a strong, diffuse, and finely granular pattern of cytoplasms.

**Figure 3 medicina-58-00900-f003:**
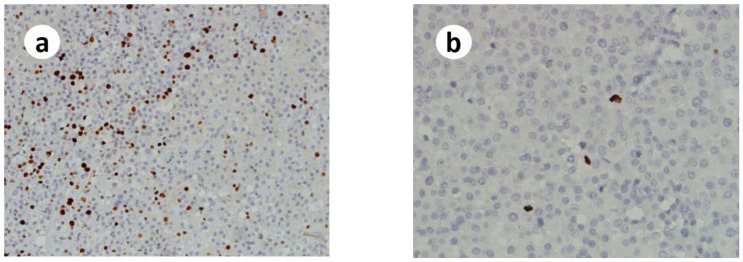
(**a**) A “hotspot” of the Ki67 proliferative index is approximately 10%. (**b**) A “hotspot” of the PHH3 immunoexpression of this case.

**Table 1 medicina-58-00900-t001:** A summary of the clinicopathologic features of the metastatic cases described up to now including the case described in this report.

	Case 1 [4]	Case 2 [5]	Case 3 [6]	Case 4 [1]	Case 5 [1]	Case 6 [7]	Case 7 [8]	Current Case
**Age (years)**	74	54	54	53	41	45	45	42
**Sex**	F	M	M	F	F	F	M	F
**Laterality**	L	R	R	L	L	R	L	L
**Metastatic site**	ovary	Lung, rib, liver, contralateral adrenal gland	Lung, right hip	Not stated	Not stated	Femur, liver, lung, contralateral adrenal gland	lung	Scalp, lung
**Clinical presentation**	Not stated	Not stated	Not stated	Virilization	Virilization	Abdominal pain	abdominal bloating	Abdominal pain
**Serum biochemistry**	Not stated	↑Cortisol	Not stated	↑Androg	↑Androg	Not stated	Not stated	Normal
**Size (cm); weight (g)**	Not stated	Not stated	Not stated	13; 670	28.5; 5720	11; 410	24; 6500	3.9;
**Mitotic count per 50HPF**	Not stated	14	Not stated	10	5	>5	6	<5
**Atypical mitotic figures**	Not stated	Absent	Not stated	Present	Present	Not stated	Present	Absent
**Venous invasion**	Not stated	Present	Not stated	Present	Absent	Not stated	Absent	Absent
**Microscopic necrosis**	Not stated	Present	Not stated	Present	Present	Not stated	Present	Present
**Capsular invasion**	Not stated	Absent	Not stated	Present	Present	Absent	Absent	Absent
**Sinusoidal invasion**	Not stated	Absent	Not stated	Present	Present	Not stated	Not stated	Present

Abbreviations: F, female; M, male; L, left side; R, right side; Androg., androgens; ↑, increased.

## Data Availability

Not applicable.

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
