# Peer review of "A Case of Oncocytic Adrenal Cortical Neoplasm with Uncertain Malignant Potential Turned Out to Be Oncocytic Adrenal Cortical Carcinoma with Distant Metastasis: Could Pathology Do Better Initially?"

_medicina, 2022, doi:10.3390/medicina58070900_

Round 1
Reviewer 1 Report
This work from Chien-Peng Huang et al. describes “a case of oncocytic adrenal cortical neoplasm with uncertain malignant potential turned out to be oncocytic adrenal cortical carcinoma with distant metastasis”. Authors raise awareness of useful observations for differential diagnosis between oncocytic adrenocortical neoplasm with uncertain malignant potential and oncocytic adrenal carcinoma.
The following comments should be taken into consideration:
Authors should include photos taken from a “hot spot” of the PHH3 immunoexpression which is the strongest indication of the initial tumor malignancy.
Secondly, additional photos should be provided from a “hot spot” of the Ki67 immunoexpression. With this demonstration, they will be able to prove the “Helsinki score” of the case reported.
Author Response
Response 1: We’ve taken photos from a “hot spot” of the PHH3 immunoexpression and gained the mitotic figure more than 5 mitoses per 50 high power fields in one of these hotspots, which met one of the major criteria of Lin-Weiss-Bisceglia histological system for diagnosing oncocytic ACC.

Reviewer 2 Report
Huang et al. reported a case of oncocytic adrenal cortical carcinoma with distant metastasis initially diagnosed as oncocytic adrenal cortical neoplasm.
This neoplasm is ultra-rare (only 56 cases described until now) and the authors added data to the shortage of literature about the diagnostic approach and histopathological characteristics.
I suggest to insert a table with the clinical and histopathological features of the metastatic cases described until now including the current one.
Author Response
Please see the attachment.

This manuscript is a resubmission of an earlier submission. The following is a list of the peer review reports and author responses from that submission.